# Queen Caging and Oxalic Acid Treatment: Combined Effect on Vitellogenin Content and Enzyme Activities in the First Post-Treatment Workers and Drones, *Apis mellifera* L.

**DOI:** 10.3390/ani12223121

**Published:** 2022-11-12

**Authors:** Simona Sagona, Francesca Coppola, Antonio Nanetti, Ilaria Cardaio, Elena Tafi, Lionella Palego, Laura Betti, Gino Giannaccini, Antonio Felicioli

**Affiliations:** 1Department of Veterinary Sciences, Pisa University, Viale delle Piagge 2, 56124 Pisa, Italy; 2Department of Pharmacy, Pisa University, Via Bonanno 6, 56126 Pisa, Italy; 3CREA Research Centre for Agriculture and Environment, Via di Corticella 133, 40128 Bologna, Italy; 4Department of Clinical and Experimental Medicine, Pisa University, via Savi 10, 56126 Pisa, Italy

**Keywords:** vitellogenin, glucose oxidase, phenoloxidase, antioxidant enzymes, *Varroa destructor*, *Apis mellifera*, oxalic acid, queen caging

## Abstract

**Simple Summary:**

Varroa destructor is a mite causing colony collapse in *Apis mellifera*. Common solutions for beekeepers to control *Varroa* mites are drone brood removal and queen caging and/or chemical treatments with formic or oxalic acid. Treatments performed against Varroa mites may affect honey bee welfare; they have the potential to cause negative effects on the immune system, as well as oxidative stress. In this study, effects of the combination of queen caging and oxalic acid treatment on both the immune system and antioxidant enzymes of first post-treatment generation workers and drones are investigated. The combination of the above anti-varroa treatments did not produce significant effects on the antioxidant system of the first post-treatment generation. However, within the immune system, such treatments determined a decrease in glucose oxidase activity in drones, and an age-dependent variation in vitellogenin content in worker bees. Such effects may result in cuticular sclerotization issue, dehydration, and pathogens transmission in drones, and in a general weakness of the immune system of both drones and workers with a subsequent higher risk of illnesses. Further investigations to assess the physiologic effects of such enzymatic activity variation on the welfare of honey bees subject to queen caging and oxalic acid treatment are desirable.

**Abstract:**

*Varroa destructor* is a mite causing serious damage to western honey bees. Managed colonies require artificial varroa control, which may be best obtained by combining mechanical and chemical methods. This study explored the possible effects of the combination of queen caging and oxalic acid treatment on the immune system (glucose oxidase, phenoloxidase, and vitellogenin) and antioxidant enzymes (superoxide dismutase, catalase, and glutathione S transferase) of first post-treatment generation drones and workers (newly emerged, nurses, and foragers). The combination of queen caging and oxalic acid treatment caused a decrease in glucose oxidase activity only in drones. This could cause issues of cuticular sclerotization, making a drone prone to bite injuries, dehydration, and pathogens. No differences in phenoloxidase activity were recorded in both post-treatment drones and workers generation. Among worker bees, the treatment determined a lower vitellogenin content in newly emerged bees while the result was higher in nurse bees. However, the treatment did not significantly affect the antioxidant enzymes activity in either drones or workers. The results obtained in this investigation suggest that the combined anti-varroa treatments had no negative effects on oxidative stress in the first post-treatment generation bees, while effects did occur on the immune system. Further investigations on the potential effects of glucose oxidase decrease in drones and vitellogenin content variation in workers are desirable.

## 1. Introduction

*Varroa destructor* is a mite causing serious damage to western honey bees, *Apis mellifera* [1]. It contributes to colony collapse, being a vector of viruses (e.g., Kashmir bee virus, Sacbrood virus, Acute bee paralysis virus, Israeli acute paralysis virus, and deformed wing virus), and feeding on bee body fat [1,2]. The parasite’s life cycle is closely intersected with that of the host. Female mites have two distinct stages: a so-called ‘phoretic’ phase, in which they live on the adult bees, and a reproductive phase, which occurs within the sealed brood cells [3].

Varroa mites can be controlled with a combination of beekeeping practices and chemical treatments. Drone brood removal and queen caging are common beekeeping practices for varroa control [4]. Many chemical compounds have also been tested over the years and formic acid, thymol, and oxalic acid are the most successful chemical treatments used at present [5]. Oxalic acid is often applied by trickling or by sublimation and its efficacy is reported to depend on both environmental conditions and colony development [6,7]. A solution of oxalic acid and sucrose applied by trickling appears to be highly effective in summertime against phoretic mites [8,9]. Oxalic acid kills Varroa through contact; however, it does not penetrate the brood seals, making it unable to reach the mites during their reproductive phase [3]. Thus, frequent periodic applications of oxalic acid during brood-rearing periods are unable to bring *V. destructor* populations below treatment thresholds [6,10]. For this reason, to maximize the oxalic acid efficacy against Varroa mites, the treatment needs to be carried out on broodless colonies or, at least, when only unsealed brood cells are present [7,8]. Queen caging in combination with acaricide applications can increase the efficacy of varroa control to more than 96% [11,12,13]. Gregorc and colleagues [4] found that the application of oxalic acid, which is non-toxic to the adult bees, in combination with queen caging or brood removal, can ensure adequate varroa control prior to the main spring honey production flow.

Both Varroa mites and varroa treatment may affect honey bee welfare, specifically via the immune system and oxidative stress. Innate honey bee immunity includes both social and individual immunity systems [14]. Glucose oxidase is an enzyme belonging to the social immune system and is responsible for the conversion of glucose into gluconic acid and hydrogen peroxide, the latter providing high antimicrobial activity [15]. Glucose oxidase is produced by hypopharyngeal glands, but its activity was observed also in salivary glands [16,17,18]. The enzyme phenoloxidase is part of the individual immune system and, through melanin production, is responsible for pathogen encapsulation and nodule formation [19]. Vitellogenin plays a pivotal role in individual immunity, as it binds to pathogen-associated pattern molecules (e.g., lipopolysaccharides, peptidoglycan, and yeast zymosan), providing the hemocytes with the zinc needed for their immune function [20,21]. Vitellogenin also regulates honey bee aging by its oxidation potential, and its gene expression is upregulated in injured bees [21]. The antioxidant system of honey bees serves to buffer the effects of oxidative stress and mainly consists of three enzymes: superoxide dismutase (SOD), catalase, and glutathione S transferase [22,23]. These three enzymes can eliminate the free radicals produced in physiological, pathological, and stressful conditions [24,25,26].

The aim of this study was to evaluate the effects of a combination of queen caging and oxalic acid treatment on both the immune system (glucose oxidase and phenoloxidase activity, vitellogenin content) and the antioxidant enzymes (SOD, catalase, and glutathione S transferase activity) of the first post-treatment generation workers and drones. 

## 2. Materials and Methods

### 2.1. Bee Collection and Haemolymph Sampling 

This investigation was performed in May 2018 and included 20 colonies from the same apiary. Ten colonies with caged queens functioned as the experimental group. The remaining 10 colonies, with uncaged queens, were left untreated and unmanipulated and functioned as the control group. 

The queens of the Experimental Group were confined for 22 days in a commercial Varcontrol cage (API-MO.BRU, Campodoro, Padua, Italy) measuring 5 × 7.8 × 3 cm. The cage was inserted in the upper part of one of the central combs of each colony. Queen-tending was made possible by the presence of queen excluders on both cage walls. On the 22nd day, the queens were released to allow egg-laying and the experimental colonies were treated with Api-Bioxal (Chemicals Laif S.p.A., Padua, Italy) (i.e.: oxalic acid dihydrate) by trickling following the label instructions (Figure 1).

From both groups (control and experimental group), honey bees were collected for enzymatic analysis and stored at −20 °C. Honey bees collected were: i. 20 newly emerged workers, approximately 21−28 days after queen release; ii. 20 drones (newly emerged) approximately 24−31 days after queen release; iii. 20 nurse bees, approximately 30 days after queen release; iv. 20 foragers, approximately 40 days after queen release. None of the experimental group individuals sampled had any direct contact with oxalic acid and the queen was present during their embryogenesis.

Twenty pools of haemolymph for each sex/age/treatment were also collected from the same colonies for a total of 160 pools. Each pool included 10 honey bees and was considered as a replicate. Each bee was narcotized with CO_2_ and 3 µL of haemolymph was withdrawn from the thorax by insertion of a 1 µL glass microcapillary through the neck membrane. Haemolymphs collected from each bee of each pool were directly put in a 1.5 mL tube, centrifuged at 3000 rpm for 20 min and then stored in PBS (80 µL of PBS × 10 µL of haemolymph) at −20 °C until analysis [27].

Spectrophotometric and colorimetric analyses were performed by an EnSpire 2300 Multilabel Reader (PerkinElmer, Milan, Italy), a Multiskan FC reader (Thermo Scientific, Waltham, MA, USA), and a Lambda 25 UV/VIS spectrometer (PerkinElmer, Milan, Italy). All chemicals were from Sigma (St. Louis, MO, USA).

### 2.2. Immune System Enzymes

The bee head was used for glucose oxidase analyses. For each sex/age/treatment, 20 heads were separately analysed. Each head was weighed before protein extraction. Then 300 µL of 100 mM phosphate buffer pH 7.2 with 1% (*v*/*v*) Triton X-100 were added. Sample was homogenized with a Teflon pestle and allowed to decant. The resulting supernatant was collected, whereas 200 µL of 100 mM phosphate buffer pH 7.2 were added to pellets and allowed to decant. The supernatant was mixed with those previously collected and the total protein concentration was measured by Qubit 2.0 fluorimeter (Invitrogen, Waltham, MA, USA). 

Glucose oxidase was then measured according to Sagona and colleagues [28]. Before analysis, a solution containing 100 mM Hepes buffer pH 7.0, 0.1 mM EDTA and 5 mM D-glucose was added to the sample. Absorbance data were obtained at the λ = 352 nm, at times 0 and 120 min, after the addition of diaminobenzidine (DAB) (0.18 mg/mL) and HRP (horseradish peroxidase) (0.02 mg/mL). Values were expressed as U/mg of proteins.

Phenoloxidase activity was investigated on 80 pooled haemolymph samples (10 pools for each sex/age/treatment). Fifty µL of each sample were loaded in cuvettes with 475 µL of phosphate saline buffer pH 7.4 and 675 µL milliQ water, following Mazzei and colleagues [29] protocol. Cuvettes were incubated at 37 °C for 5 min and 300 µL L-3,4-dihydroxyphenylalanine (L-dopa) (2 mg/mL) was then added. Absorbance data were obtained at λ = 490 nm, at time 0 and 10 min. Values were expressed as U/mg of proteins.

### 2.3. Vitellogenin Assay Kit

Vitellogenin content was measured in all the 160 pooled haemolymph samples (20 pools for each sex/age/treatment) using honey bee vitellogenin (VG) ELISA Kit (MyBiosource, San Diego, CA, USA), according to the manufacturer’s instructions. Fifty μL of each sample diluted 1:20, two blanks with a sample dilution solution, and 6 standards with a known concentration of vitellogenin were loaded into a 96-well plate. Optical density was recorded at the λ = 450nm. The average of the two blanks was detracted from the data obtained and the result was fitted to the calibration curve (obtained with the standards) using the MyCurveFit.com (accessed on 20 July 2018) program, obtaining the corresponding ng/ml vitellogenin for each sample. 

### 2.4. Antioxidant Enzymes

Antioxidant enzymes were investigated in 80 pooled haemolymph samples (10 pools for each sex/age/treatment) in accordance with Sagona and colleagues [23].

SOD activity was determined using 10 μL of each sample by Superoxide Dismutase Assay kit (Cayman Chemical Company, Ann Arbor, Michigan, USA; No.706002). This kit uses a tetrazolium salt for the detection of superoxide radicals generated by xanthine oxidase and hypoxanthine and measures the three types of SOD (Cu/ZnSOD, MnSOD, and FeSOD). One unit of SOD is defined as the amount of enzyme needed to exhibit 50% dismutation of the superoxide radical. The kit was used according to the manufacturer’s instructions and the plates were read at the λ = 450 nm.

Catalase activity was determined using 20 μL of each sample by Catalase Assay kit (Cayman Chemical Company, Ann Arbor, MI, USA; No.707002). This method is based on the reaction of the enzyme with methanol in the presence of an optimal concentration of hydrogen peroxide. The formaldehyde produced is measured colorimetrically with 4-amino-3-hydrazino-5-mercapto-1,2,4-triazole (Purpald) as the chromogen. The kit was used according to the manufacturer’s instructions and the plates were read at the λ = 540 nm.

Glutathione S-transferase was determined using 20 μL of each sample by Glutathione S-transferase Assay kit (Cayman Chemical Company, Ann Arbor, MI, USA; No.703302). This kit measures the total GST activity (cytosolic and microsomal) by measuring the conjugation of 1-chloro-2,4-dinitrobenzene (CDNB) with reduced glutathione in terms of increased absorbance at 340 nm. The kit was used according to the manufacturer’s instructions and the plates were read at the λ = 340 nm for 5 min.

### 2.5. Statistical Analysis

Data were statistically processed using JMP software (SAS Institute, Cary, NC, USA, 2008). The vitellogenin content and all enzymatic activities (except for glutathione S transferase activity) were processed as follows. After assessing that the data distribution was significantly (*p* < 0.05) different from normal using a Shapiro–Wilk test, a non-parametric Wilcoxon test was adopted, with the group as the independent predicting variable. As glutathione S transferase activity was normally distributed, data were processed by ANOVA test followed by a Student’s *t*-test. Data were analysed within each age/sex and between workers of different ages. Differences associated with *p* < 0.05 were considered statistically significant.

## 3. Results

### 3.1. Immune System Enzymes Activity and Vitellogenin Content

Glucose oxidase activity was lower among drones in the treated colonies compared to the control (202 ± 31 vs. 374 ± 38 U/mg of proteins, respectively, *p* = 0.0011), while no significant differences were found in the three worker bees age investigated (Figure 2). 

No significant differences were detected in phenoloxidase activity in both workers and drones (Table 1). Vitellogenin content in the heamolymph of newly emerged bees was significantly higher in control colonies than in the experimental group (2047 ± 454 vs. 894 ± 214, *p* = 0.0412). In nurse bees, vitellogenin content resulted significantly lower in the control than in the experimental group (288 ± 68 vs. 650 ± 109, *p* = 0.0303). Conversely, no significant variation in vitellogenin content was recorded in foragers and drones (Figure 3).

### 3.2. Antioxidant Enzymes Activity

No significant differences in antioxidant activity of SOD, catalase, and glutathione S transferase were observed in both worker bees and drones belonging to control and experimental colonies (Table 2).

## 4. Discussion

Glucose oxidase activity shows no significant variation in treated worker bees from each age group compared to control bees. Glucose oxidase activity was also detected in both treated and control drones. This result clashes with the common assumption that drones do not produce glucose oxidase since they do not have hypopharyngeal glands [30,31], but it agrees with Kairo et al., [32] who measured glucose oxidase in the head of drones. This enzyme was also detected in drone proteome during embryogenesis [33]. Despite the lack of hypopharyngeal glands, the drones have post-cerebral glands (delicate masses of tiny follicles), rudimentary mandibular glands, and salivary glands, the latter being similar to those of the females [30]. Therefore, the detection of glucose oxidase activity in the head of drones, recorded in this study, suggests that this enzyme might be produced by post-cerebral glands. In addition, the drones have a group of fat cells provided with ducts that open in the front end of the esophagus [34]. The function of these cells is currently unknown however their involvement in the production of glucose oxidase cannot be excluded. Results obtained in this study also show a lower glucose oxidase activity in the drones from treated colonies compared to the control ones. An increase in glucose oxidase activity associated with a decrease in survival rate has been recorded in foragers under stress conditions [28]. Therefore, the lower glucose oxidase activity detected in treated drones in this study could indicate a low impact of treatment on their general welfare. Since the reason why drones produce glucose oxidase is still unknown, further investigations are desirable. 

Glucose oxidase was also identified in the cuticle and haemolymph of locusts [35]. It has been hypothesized that in locusts, this enzyme acts as an adjuvant of cuticle sclerotization, whereas its presence in the haemolymph has not yet been clarified [35]. A possible role of glucose oxidase in cuticular sclerotization also in drones might be speculated and needs to be deeply investigated, since cuticular sclerotization issues could make a drone prone to bite injuries, dehydration and pathogens. 

In both workers and drones, phenoloxidase activity was not significantly affected by the experimental treatment. As phenoloxidase belongs to the individual innate immune system of honey bees, it is not surprising that individuals from healthy colonies did not show altered patterns in the activity of this enzyme [19,29,36]. 

Vitellogenin in drones has already been investigated [37]. The low vitellogenin content found in drones in this investigation agrees with Bitondi and Simoes [38], who hypothesized a relationship between the synthesis of new vitellogenin and pollen consumption, as drones eat less pollen than workers [39].

Data obtained in this investigation indicate an age-dependent effect of vitellogenin content in workers. Specifically, the treatment determined a significantly lower vitellogenin content in newly emerged bees and a higher vitellogenin content in nurse bees compared to control. Cabbri and colleagues [40] also recorded an increase in vitellogenin content in worker bees from colonies treated with oxalic acid and with caged queens. Vitellogenin is suggested to be involved in brood food production which is produced by nurse bees [41]. Therefore, nurse bees produce and consume vitellogenin, while in old bees vitellogenin is partially accumulated in the organism [41]. The larval instars were tended by nurse bees [42]. In this study, experimental colonies nurse bees tending larvae of the post-treatment generation were exposed to oxalic acid. Therefore, differences in vitellogenin content recorded in workers may be related to the care received during their larval development. The exposure of tending nurses to oxalic acid may have caused the decrease in vitellogenin content in the newly emerged post-treatment generation; thus, the increase in vitellogenin content recorded in the nurse bees post-treatment generation may be driven by a need to compensate for the initial low content towards royal jelly production. Considering all the potential immunological and antioxidant effects of vitellogenin, it can be hypothesized that a decrease in vitellogenin content below a physiological threshold may negatively impact honey bee welfare. All three antioxidant enzymes investigated (SOD, catalase, and glutathione S transferase) were not significantly affected by the anti-varroa combined treatment. This suggests that the generation following the treatment did not suffer drastic effects in terms of oxidative stress. 

## 5. Conclusions

In conclusion, the results obtained in this study provide an overview on the effects of the anti-varroa treatment (i.e., queen caging and oxalic acid treatment) on the first post-treatment generation of workers and drones. The results obtained in this investigation suggest that the combined treatment had no negative effect on oxidative stress, while an effect on the immune response of the post-treatment bee generation occurred. Combined anti-varroa treatment caused a decrease in glucose oxidase in drones and an age-dependent variation in vitellogenin content in worker bees, with potential impact on bee welfare resulting from a general weakness of the bee immune system and subsequent higher risk of illness. Further investigations of the potential consequence of the decrease in glucose oxidase in drones and of the variation in vitellogenin concentration in different worker ages are desirable. In addition, further investigations to assess the physiologic effects of enzymatic activity variation impacting the welfare of honey bees subject to queen caging and oxalic acid treatment are also desirable. 

## Figures and Tables

**Figure 1 animals-12-03121-f001:**
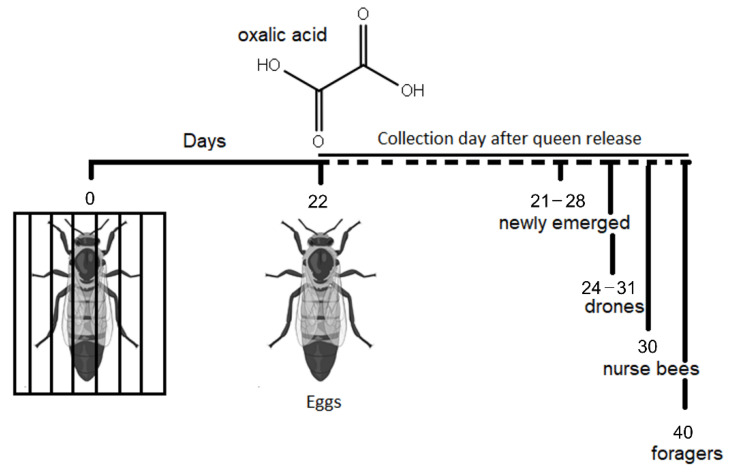
Experimental design for sample collection. Figure created with Biorender.com, accessed on 17 March 2022.

**Figure 2 animals-12-03121-f002:**
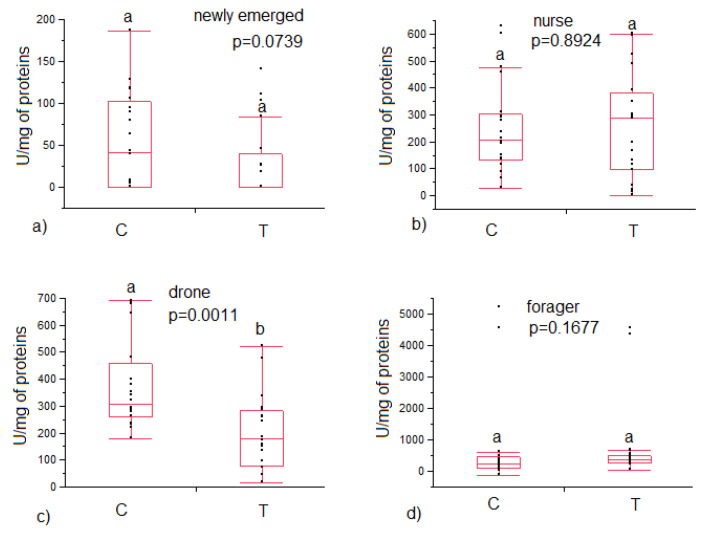
Glucose oxidase activity in (**a**) newly emerged, (**b**) nurse, (**c**) drone and (**d**) foragers belonging to control (C) and treated (T) colonies. Statistical differences are indicated with different letters.

**Figure 3 animals-12-03121-f003:**
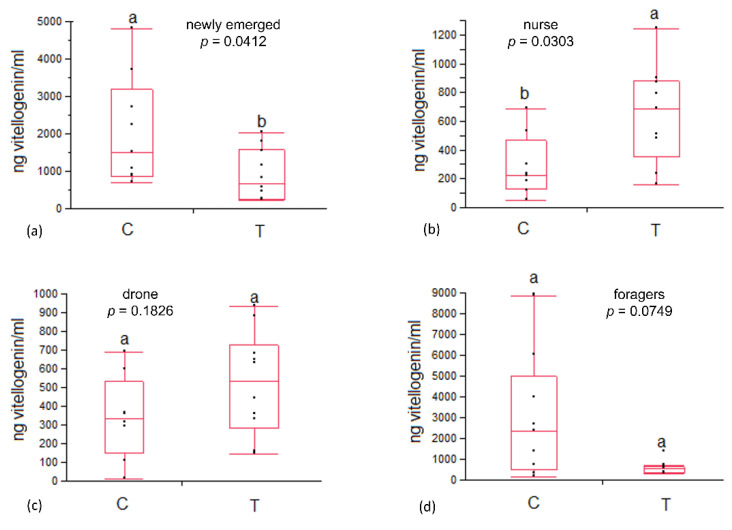
Vitellogenin content in (**a**) newly emerged, (**b**) nurse, (**c**) drone and (**d**) foragers belonging to control (C) and treated (T) colonies. Statistical differences are indicated with different letters.

**Table 1 animals-12-03121-t001:** Phenoloxidase activity in honey bees belonging to control and experimental colonies (queen caging + acid oxalic). Values are reported as mean±SE (median).

	Control Colonies	Experimental Colonies	*p* Value	Test Value
Phenoloxidase activity U/mg of proteins				
Newly emerged bees	124 ± 43(129)	236 ± 66(138)	0.3607	χ^2^ = 0.8354, df = 1
Nurse bees	350 ± 111(264)	205 ± 51(234)	0.5939	χ^2^ = 0.2843, df = 1
Foragers	248 ± 82(205)	144 ± 43(116)	0.3250	χ^2^ = 0.9686, df = 1
Drones	310 ± 108(260)	172 ± 48(136)	0.3073	χ^2^ = 1.0422, df = 1

**Table 2 animals-12-03121-t002:** Antioxidant enzymes of honey bees belonging to control and experimental colonies (queen caging + oxalic acid): SOD activity (U/mg of proteins, n = 10); catalase activity (nmol/min/mg of proteins n = 10); glutathione S transferase activity (nmol/min/mg of proteins). Values are reported as mean ± SE (median).

	Control Colonies	Experimental Colonies	*p* Value	Test Value
SOD activity U/mg of proteins				
Newly emerged bees	3.96 ± 0.36 (3.78)	4.23 ± 0.42 (4.52)	0.7624	χ^2^ = 0.0914, df = 1
Nurse bees	4.03 ± 0.31 (4.13)	4.49 ± 0.21 (4.46)	0.4905	χ^2^ = 0.4754, df = 1
Foragers	5.08 ± 0.33 (5.53)	5.37 ± 0.30 (5.58)	0.4057	χ^2^ = 0.6914, df = 1
Drones	4.29 ± 0.43 (4.39)	3.52 ± 0.35 (3.48)	0.5453	χ^2^ = 0.3657, df = 1
Catalase activity nmol/min/mg of proteins				
Newly emerged bees	16.66 ± 3.85 (13.79)	12.08 ± 3.88 (9.76)	0.2263	χ^2^ = 1.4640, df = 1
Nurse bees	56.89 ± 4.32 (55.40)	52.07 ± 7.31 (47.37)	0.4497	χ^2^ = 0.5714, df = 1
Foragers	15.82 ± 3.39 (13.24)	19.75 ± 3.03 (22.34)	0.2568	χ^2^ = 1.2857, df = 1
Drones	10.20 ± 3.97 (5.04)	7.37 ± 1.27 (7.51)	0.8206	χ^2^ = 0.0514, df = 1
Glutathione S transferase activity nmol/min/mg of proteins				
Newly emerged bees	161 ± 26 (142)	128 ± 24 (136)	0.5576	F = 0.3563, df = 1
Nurse bees	145 ± 17 (138)	197 ± 21 (180)	0.0569	F = 4.0394, df = 1
Foragers	156 ± 19 (159)	166 ± 16 (171)	0.6793	F = 0.1760, df = 1
Drones	106 ± 16 (116)	148 ± 19 (131)	0.1091	F = 2.8123, df = 1

## Data Availability

The data used to support the findings of this study can be made available by the corresponding author upon request.

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
