# Peer review of "Queen Caging and Oxalic Acid Treatment: Combined Effect on Vitellogenin Content and Enzyme Activities in the First Post-Treatment Workers and Drones, Apis mellifera L."

_animals, 2022, doi:10.3390/ani12223121_

Round 1

Reviewer 1 Report

Queen Caging  and Oxalic acid Treatment... - Sagona et al.

An interesting and useful study of a factor that should have been studied a long time ago with the introduction of oxalate treatment regimen for mites in bee colonies. The study is well executed, and the results and discussion of the results are satisfactory. Yet the authors need to fine tune their methods because as it is currently, it seems wanting for more details. The primary headline of this study is that, in drones, the treatments seem to affect the glucose oxidase titers and that could affect the cuticular sclerotization. In the case of the newly emerged bees and the nurse bees the vitellogenin levels are afflicted down and up, respectively. I think the authors must highlight these aspects in their abstract. Also, a note on how these changes would/could affect the bees. For example, cuticular sclerotization issues could make a drone prone to bite injuries, dehydration, as well as pathogens. Finally, the authors must check how their statements convey the intended message to the readers by requesting someone else to read the manuscript. Additional minor comments and suggestions are below.

Line 20: Change ’striking’ to ’significant’

Line 34: ”The activity of the three antioxidant enzymes that were assessed were not significantly affected in either the drones or the workers”.

Line 35: Just as the suggestion with respect to ’Line 20’, avoid using adjectives/adverbs that could not not be scientifically evaluated. Words such as, ’drastic’ must be changed to something else.

Line 44: modify sentence to ”... and because they feed on bee fat body[1,2].”

Line 59: italicize V. destructor

Line 64: ”...oxalic acid, which is non-toxic to the adult bees, in combination with...”

Line 67-68: Change, ”in detail” to ”specifically”

Line 111-113: Could you please explain how hemolymph was extracted (location on the bee, was a syringe or a capillary used, any additives, previous method reference, etc.)? Were the samples pooled and stored? If so, are the assays technical replicates? How many bees contributed to the pool, if pooled?

Line 141: Fifty ”l” ? Please fix this. If that is 1 (one) all single digit in the manuscript must be spelt-out.

Section Methods/Antioxidant enzymes: Each assay describes what the kit does. However, it would be more useful to note the amount of hemolymph used per assay and such details.

Results: The very first sentence could be converted to a more active voice and direct statement such as, ”Glucose oxidase activity was lower among the drones in the treated colonies compared to the control (20231 vs...”

Results: I think the results should be more descriptive with added sections reflecting the sections in the materials and methods.

Results (optional suggestion): Nothing wrong with the current form of all results as tabulated summary results. However, converting tables to graphs (either columnar with error bars or box graphs at least for the assays that show significant differences) would be more appealing and the results would also be easy to comprehend.

Line 233: change ”like in the case of this study” to ”as noted in our study”. 

Reviewer 2 Report

line 54  Treatment of oxalic acid in solution with sucrose by trickling can determine exhibit high efficacy against the phoretic mites during in summer

line 141 . Fifty l of each sample, two blanks with a sample dilution solution, is this a typo?  should it be microliters?

line 260 . In spring oxalic acid is depleted shortly [42] This citation does not apply, since the temperatures were far higher than in a hive, and the atmospheric pressure is lower.  My own data (in prep) indicate that some oxalic acid residues persist on the bees in a hive for over a month.

Overall a great study -- well designed, with clear experimental methods detailed, and sober conclusions.  The differences in Vg are of great interest, perhaps more so than you concluded.
